# A measure of event-related potentials (ERP) indices of motivation during cycling

Rémi Renoud-Grappin[1], Damien Gabriel [1,2,3]*, Emmanuelle Broussard[2], Laurent Mourot [4,5], Julie Giustiniani[1,2,6], Lionel Pazart [1,2,3]

1 Université Marie et Louis Pasteur, INSERM, UMR 1322 LINC, Besançon, France, 2 CHU Besançon, Inserm CIC 1431, Besançon, France, 3 Plateforme de Neuroimagerie fonctionnelle et neuromodulation Neuraxess, Besançon, France, 4 Université de Franche-Comté, SINERGIES, Besançon, France, 5 Department of Biological Sciences, Faculty of Science, Thompson Rivers University, Kamloops, Canada, 6 Service d'addictologie, Centre Hospitalier Universitaire, Besançon, France

* dgabriel@chu-besancon.fr

## Abstract

Although motivation is a central aspect of the practice of a physical activity, it is a challenging endeavour to predict an individual's level of motivation during the activity. The objective of this study was to assess the feasibility of measuring motivation through brain recording methods during physical activity, with a specific focus on cycling. The experiment employed the Effort Expenditure for Reward Task (EEfRT), a decision-making task based on effort and reward, conducted under two conditions: one involving cycling on an ergometer at moderate intensity and the other without cycling. The P300, an event-related potential linked to motivation, was recorded using electroencephalography. A total of 20 participants were recruited to complete the EEfRT, which involved making effort-based decisions of increasing difficulty in order to receive varying levels of monetary reward. The results demonstrated that the P300 amplitude was influenced by the act of cycling, exhibiting a reduction during the cycling session. This reduction may be explained by a reallocation of cognitive resources due to the exertion of physical effort, which is consistent with the transient hypofrontality theory. In terms of behaviour, participants demonstrated a tendency to make more challenging choices when the potential rewards were higher or the probability of gaining them was lower. This pattern was observed in both the cycling and non-cycling conditions. A positive correlation was identified between P300 amplitude and the proportion of difficult choices, particularly under conditions of low reward probability. This suggests that P300 may serve as a neural marker of motivation. The study demonstrates the feasibility of using electroencephalography to monitor motivation during exercise in real-time, with potential applications in rehabilitation settings. However, further research is required to refine the design and explore the effects of different exercise types on motivation.

**Data availability statement:** The data and materials necessary to reproduce the findings reported in this study are available at OSF repository, DOI 10.17605/OSF.IO/DTJXV.

**Funding:** The Conseil Régional de Bourgogne-Franche-Comté funded the doctoral grant of Rémi Renoud-Grappin. The funders had no role in study design, data collection and analysis, decision to publish, or preparation of the manuscript.

**Competing interests:** The authors have declared that no competing interests exist.

## Introduction

There is a substantial body of evidence indicating the positive effects of physical activity on various aspects of health, including physical, mental, and social well-being, as well as mortality. Regular physical activity has been identified as a key factor in maintaining physical and mental health, as well as in preventing a range of diseases, including neurological, cardiovascular and diabetes. However, according to the WHO ("Physical activity" 2024), approximately one-third of adults do not engage in the recommended level of physical activity. Consequently, a 15% reduction in physical inactivity by 2030 compared to 2010 has been targeted.

A significant obstacle to physical inactivity is motivation, or more specifically, the lack thereof [1]. The concept of motivation is central to the understanding of human behaviour. In theory of self-determination [2], a number of distinct forms of motivation are identified. Such actions are deemed to be self-determined or autonomous, as they are assumed to originate from the individual in question. In contrast, behaviour that is compelled or stimulated by external incentives represents the opposite end of the spectrum. The most self-determined form of motivation is intrinsic motivation, which is driven by the desire for pleasure. This is followed by integrated regulation, which is aligned with one's personal values. The third most self-determined form of motivation is identified motivation, which is driven by the recognition of the benefits associated with a particular action. Non-self-determined regulations may be classified as introjected (out of guilt), external (for rewards or praise). These regulatory types are associated with motivations and behaviours that are contingent upon external incentives, including physiological drives such as hunger, the need to alleviate pain, the influence of others, financial considerations, and so forth. At the opposite end of the spectrum, there are also non-self-determined regulations, such as amotivation, which is defined as a lack of intention to act. Despite the criticism levied against the existence of such a continuum, there is a positive correlation between self-determined forms of motivation for physical activity and intensity, duration of practice, and psychological well-being [3].

In order to encourage physical exercise, it is therefore essential to be able to predict an individual's level of motivation not only before the commencement of an exercise regimen, but also during the actual practice of that exercise. The ability to ascertain the level of motivation in real time during exercise would facilitate the capacity to modify one's behaviour and adapt the exercise accordingly in order to pursue the task. Nevertheless, there is currently no consensus on the most effective method for measuring an individual's motivational level during exercise [4]. Motivation can be quantified through the utilisation of self-report scales [5], as these scales are straightforward to administer, can be distributed to a multitude of individuals simultaneously, and are capable of targeting highly specific information. However, there can be a discrepancy between self-reported and actual behavioural outcomes [6]. Other studies employ heart rate variability measurements recorded daily and compared with self-reported fatigue or motivation responses [7]. However, heart rate variability is also dependent on the intensity and the type of exercise [8].

In this context, electroencephalography (EEG) represents a potentially fruitful avenue for directly measuring motivation at the brain level. EEG was the inaugural non-invasive methodology for the direct measurement of cerebral activity, a century ago [9]. EEG remains a widely used method, capturing the micro-currents resulting from electrical activity generated by ionic flows as neurons connect and fire [10]. One advantage of EEG over other techniques for measuring brain activity is its temporal resolution, which allows for the measurement of brain responses with a temporal precision of one millisecond. Given that motivation is frequently conceptualised within the field of cognitive neuroscience as the neural representation that predicts goal-oriented behaviours and is associated with anticipated outcomes, EEG represents a potentially optimal method for recording these neural responses. Indeed, studies have demonstrated that EEG signals can be correlated with motivation during a task, particularly through the measurement of event-related potentials (ERPs) [11,12]. Event-related potentials (ERPs) are defined as changes in electrical potential produced by the nervous system in response to internal or external stimuli. Event-related potentials (ERPs) have been linked to a number of cognitive processes, including attention, inhibition, response choice, error feedback processing, memory activity and motivation [13]. Additionally, ERP measures have been linked to low-level sensory processes, such as perception. One ERP that has been linked to motivation is the P300, which is a positive response that peaks between 300 and 600 ms post-stimulus, with the largest amplitude occurring at centro-parietal scalp sites [14,15]. The amplitude of the P300 is often described as proportional to the motivational level [16–20]. A P300 related to motivation can be recorded using a motivational and decision-making task designated as the effort expenditure for reward task, or EEfRT [21]. The EEfRT is employed to examine motivation in the context of choice selections that encompass varying degrees of physical exertion in pursuit of a monetary reward [22]. The EEfRT is a key-pressing game in which participants indicate the level of physical effort they are willing to expend in order to obtain varying monetary rewards with differing probability levels for receipt. When the EEG is recorded during the task, a more negative response is observed when participants receive no reward, particularly on centroparietal electrodes. Furthermore, a significant positive correlation is identified between P300 amplitude and the proportion of difficult choices when the probability of receipt is low. This was described as the condition better suited to reflect the strength of motivation [21].

Up until this point, ERP measures related to motivation have consistently been conducted as standalone investigations, divorced from the context of physical activity. The measurement of brain activity during movement is a challenging endeavour, due to the presence of artefacts created by body movements [23], and the necessity of connecting the head to the equipment via cables. Significant advancements have been made in both signal processing and the equipment used [24]. A growing number of mobile EEG devices are entering the market, offering performance levels comparable to those of fixed devices [25]. Furthermore, certain exercise conditions are more conducive to the recording of cerebral activity. For instance, cycling on a bicycle or cycloergometer has been utilised for ERP measurements during aerobic exercise, given that the head remains relatively immobile. An increasing number of studies have employed the measurement of ERPs during cycling, in a variety of conditions pertaining to intensity, duration and the specific cycling context (including non-biking activities) [26]. In these studies, the P300 was the principal ERP measured, typically through oddball tasks, which indicate that this response can be analysed in a motivational context.

The objective of this study was to assess the feasibility of evaluating motivation by electroencephalogram (EEG) during physical activity, in this case cycling. The feasibility of measuring motivation-related ERPs during the same EEfRT task as in [21] but during cycling at a moderate intensity was evaluated. This was done with a view to its potential use in the rehabilitation context. Additionally, an ERP not related to the motivational context was measured, namely the feedback-related negativity (FRN). This was done on the basis of a binary classification of outcomes [16,27,28]. It was therefore anticipated that motivational processes during cycling would affect the P300, but not the FRN. Furthermore, given the substantial evidence for the beneficial effects of exercise on mood [29], and the established influence of mood on decision-making [30], the impact of physical exercise on the physical effort participants were willing to engage in was also evaluated.

## Methods

### Participants

A total of 20 healthy participants were included in the study (6 men and 14 women; mean age 26). In order to be included in the study, participants had to be between the ages of 18 and 45 and right-handed, as determined by the Handedness Questionnaire of Oldfield [31]. Participants received information regarding the aim and procedures of the experiment, and gave their written informed consent to participate in the study. Prior to their inclusion, the participants completed a brief questionnaire to ascertain whether they exhibited any of the following conditions: depression, asthma, or other pathologies that could potentially impair their ability to engage in physical exercise; psychiatric medication; pregnancy or breastfeeding; a history of cranial lesion; or participation in a study that precluded their enrollment in another. Since anhedonia, depression or other psychiatric conditions have been shown to impact EEfRT performance [22], individuals with a psychiatric history were excluded from this study. No remuneration was provided for participation. The protocol was approved by the regional ethics committee of Bourgogne-Franche-Comté (CERUBFC-2023-06-06-032). Participants were recruited from the 19th of January 2024 to the 30th of March 2024.

### Process of the experiment

The participants were required to complete two distinct sessions: one in which they did not engage in cycling, and one in which they did (Fig 1). The latter session was conducted on a cycloergometer. The order of the sessions was randomly assigned to each participant and both sessions took place in the afternoon. The interval between two sessions was kept as short as possible, depending on the participant's availability, in order to avoid variations in their psychological state as well as variations in the external environment (e.g., temperature). At the outset of the inaugural session, participants were required to complete the Edinburgh Handedness Inventory [31] in order to ascertain their handedness and ensure that their cerebral activity could be accurately gauged. The questionnaire comprises 10 items, with participants instructed to indicate their preferred hand by placing a checkmark in the appropriate box. Furthermore, the participants completed the International Physical Activity Questionnaire (IPAQ, 2003) to ascertain their physical activity habits. The IPAQ questionnaire assesses the amount of time spent exercising at moderate intensity, high intensity, walking, and sitting in a typical week to determine the participants' level of physical activity. Subsequently, the participants performed the calibration phase of the EEfRT.

In the "cycling session," participants were equipped with a chest belt to monitor their heart rate (HR) throughout the entire session. Initially, they remained in a recumbent position for five minutes to measure their resting HR. Subsequently, a target HR was calculated based on their age and resting HR using an estimated maximum HR. This target HR corresponded to 50% of the estimated reserve HR. The calculation was as follows:

$$HR\ max \ = \ 220 - age$$

$$HR\ reserve \ = \ HR\ max - HR\ rest$$

$$HR\ target(50\%) \ = \ HR\ rest \ + \ 0.5 \ x \ HR\ reserve$$

Subsequently, a power calibration phase was conducted to ascertain the target power output (expressed in watts as P target) for a personalised moderate exercise regimen during the experiment. The participants commenced the trial with a power output of P = 50W at 60 rpm. Thereafter, the power was increased by 10W every 1.5 or 2 minutes until the target heart rate was reached. The power at the target heart rate is designated as P target. Subsequently, the participants were equipped with the EEG device and required to maintain a power output of P target at 60 rpm.

A training session comprising ten trials of the EEfRT was conducted by the participants while pedalling, in order to ensure that they had a clear understanding of the task. Subsequently, the participants completed the Brief Mood Introspection Scale (BMIS) [32]. The BMIS is a 17-item mood adjective scale comprising 16 adjectives. Participants were required to evaluate their current emotional state by rating the degree to which 16 different adjectives described their mood on a scale of 1–4, with 1 indicating a low level and 4 indicating a high level of the respective emotion (e.g., happy, tired, gloomy, active). Subsequently, the participant completed the EEfRT. A rating of perceived exertion was conducted using the Borg scale, which ranges from 6 to 20. Participants were required to report their perceived exertion for the pedalling exercise, with 6 indicating no effort and 20 indicating maximal effort [33]. The Borg scale was completed on five occasions: initially after the training period and then at the conclusion of each block. At the end of the EEfRT, the BMIS was administered once more. The entire session lasted for a total of two hours.

In the "no-cycling session," participants were seated on the cycloergometer but did not engage in cycling. The task was identical to that of the cycling session, with the exception of the calculation of resting heart rate and target heart rate.

**EEfRT**

In order to facilitate the implementation of the EEfRT, a response pad was affixed to the cycloergometer (see Fig 2).

**Calibration.** Calibration was conducted to ascertain the maximum number of button presses that participants could perform in 7 seconds with the index finger of their dominant hand (right) and in 14 seconds with the thumb of their non-dominant hand (left). This allowed for the difficulty of the EEfRT to be tailored to the individual.

Prior to the commencement of the experiment, the participants were informed that they would receive chocolate tablets according to their final score.

**EEfRT task.** The EEfRT, adapted from [22] for ERP analysis [21] (Fig 3), aims to maximize points by completing easy or hard tasks. Task selection depends on the potential reward and the probability of receiving it.

In this study, the task included 120 trials. The easy task required 70% of the participant's calibrated maximum button presses with the right index finger within 7 seconds, awarding 1 point upon success. The hard task required 90% of the maximum presses using the left-hand auricular finger within 14 seconds, earning 1.5, 3, 4.5, or 6 points. The hard task's

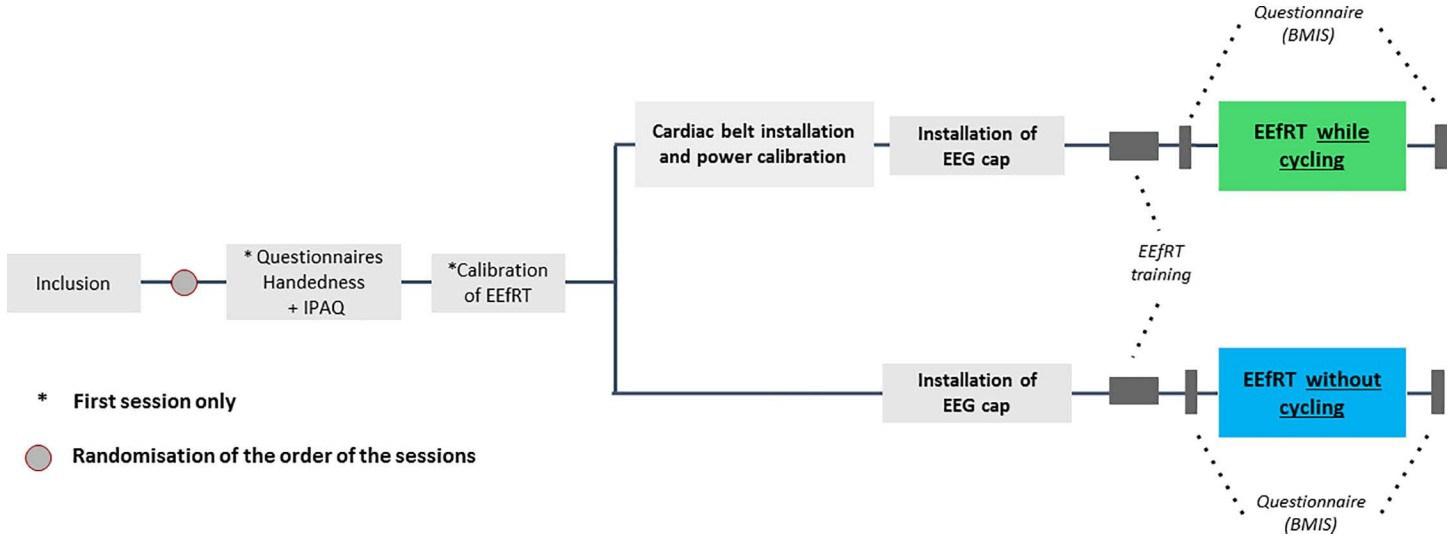

**Fig 1. Experimental protocol of the study.**

time limit was reduced from 21 to 14 seconds to accommodate the increased trial count. Reward probabilities (10%, 50%, or 90%) applied equally to both tasks.

Fig 3 outlines the trial sequence:

- A 1-second screen displayed the reward probability and point values for easy (1 point) and hard (variable points) tasks.

- A circle appeared, prompting task selection.

- Participants pressed buttons to fill a bar before time expired.

- Success was indicated by a cross (1 s) followed by feedback (2 s): a green square for a win, a red square for a loss. Failure resulted in a red square for 2 seconds.

This adaptation of the EEfRT was programmed in E-prime (Psychology Software Tools Inc.; Sharpsburg, PA, USA). The probability and amount of order was randomized across participants. To ensure task comprehension, subjects received

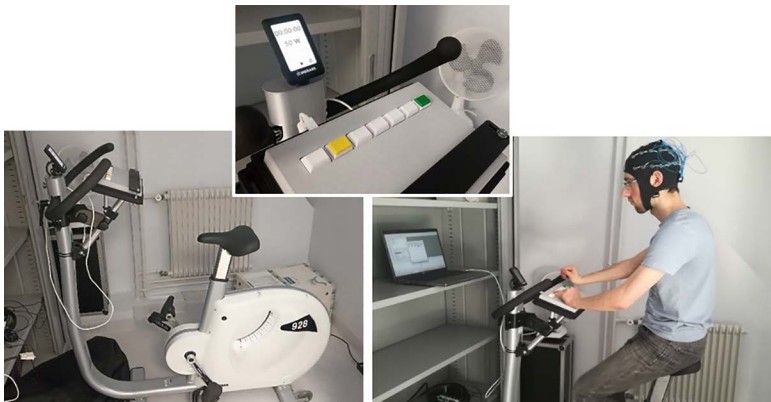

**Fig 2. Materiel used in the study.** Left: cycloergometer (Monark 928); middle: response pad; right: EEG cap (Emotiv Epoc Flex).

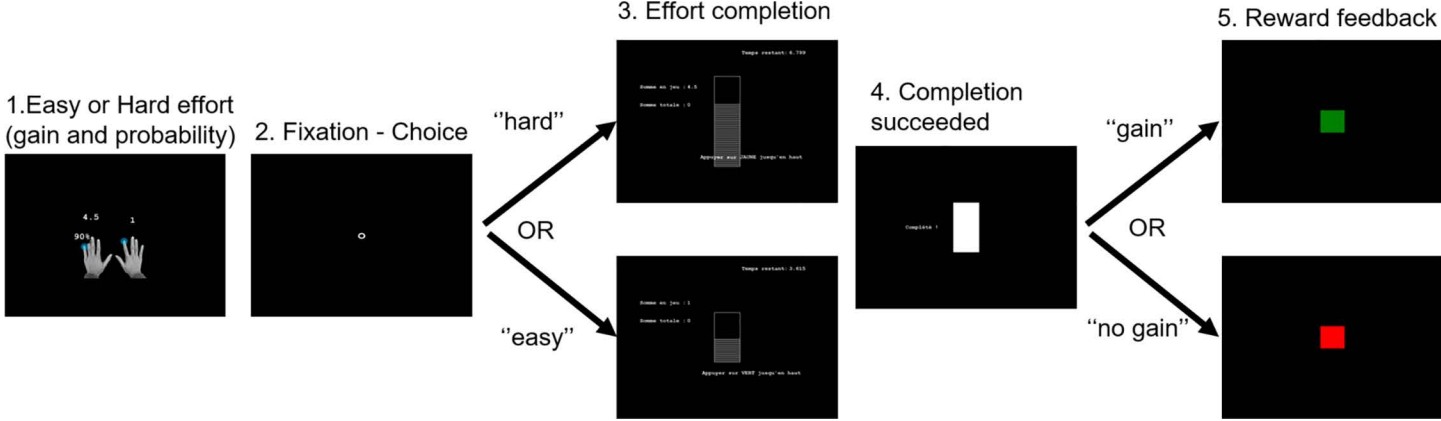

**Fig 3. Schematic diagram of a trial at the EEfRT.** 1. Presentation of the probability to receive the reward associated with the amount of the reward for easy and hard effort conditions (1s); 2. Time of selection of easy or hard effort (max 10s); 3. Completion of the effort by button press; 4. Success screen; 5. Reward feedback, the "rewarded" condition = green square and "not rewarded" condition = red square.

oral instructions and were provided with a series of task instructions. The task lasted about 35 minutes and included 3 pauses so there were 4 blocks for both sessions.

## EEG Acquisition

For both experimental sessions, participants were equipped with a 32-electrode mobile electroencephalogram (EEG) device (Emotiv Epoc Flex, https://www.emotiv.com) with the use of EEG gel. The electrode sites were distributed in an even manner across the surface of the head, and the locations were as follows: The following electrode sites were used: Fpz, Fz, Cz, Fp1, F7, F3, C3, FC3, FT7, T7, TP7, CP3, P3, P7, PO7, O1, Cpz, Pz, POz, O2, PO8, P8, P4, CP4, TP8, T8, FT8, FC4, C4, F4, F8, Fp2. The CMS electrodes were located at FCz, while the DRL electrode was situated at AFz. The sampling rate was set at 128 Hz. Stimulus occurrence was marked in the recorded EEG signal using E-Prime timestamp markers, which were displayed by a third software (LabRecorder) using Lab Streaming Layer.

## EEG processing

The data were processed using the EEGLAB software. A band-pass filter with a passband of 1–30 Hz was applied to the raw EEG signals. The Infomax ICA algorithm was employed to exclude any components related to eye blink artefacts. Trials were selected based on the timestamp markers of the reward feedback screen (green or red scare), with a window of 100 milliseconds before the feedback onset and 600 milliseconds after. The markers of feedback for wins and losses were selected on a separate basis. For each trial, a baseline correction was performed between 100 ms prior to the onset of the feedback and the onset of the feedback itself. To remove artifacts caused by movement, and any other non-physiological factors, EEG epochs were cleaned with a visual trial-by-trial inspection. For each subject, four distinct data files representing the cerebral response to the reward feedback within four distinct conditions were obtained: cycling and rewards, cycling and no rewards, no cycling and rewards, and no cycling and no rewards. In all four conditions, a mean of 30 trials were conserved for analysis.

## Data analysis

**Electroencephalogram (EEG).** For each subject and experimental condition, the data from all trials were averaged to create a single mean event-related potential (ERP) waveform. Subsequently, the mean ERPs of all subjects were averaged to create a grand-averaged ERP for each condition and electrode. Based on the grand-averaged responses, the amplitude of the ERP was calculated as the mean value over the time window 153–216 ms for the feedback-related negativity (FRN) and 223–568 ms for the P300.

The EEG data were analysed using the software Statistica with the objective of identifying significant differences in P300 and FRN amplitude among the variables. A total of nine electrodes were selected for analysis, which were divided into two categories: laterality and anteriority. The left electrodes were F3, C3 and P3. The midline electrodes were Fz, Cz, and Pz. The right electrodes were F4, C4, and P4. The anterior electrodes were F3, Fz and F4. The central electrodes were C3, Cz, and C4, while the posterior electrodes were P3, Pz, and P4. The statistical analysis of the EEG data included the following factors: outcome (reward, no reward), exercise (cycling, no cycling), laterality (left, middle, right), anteriority (anterior, central, posterior) and order (cycling and no cycling, no cycling and cycling). When a significant result was obtained, a post hoc test was conducted with a Bonferroni correction.

**Behavioral data.** The behavioural analysis of the EEfRT included an investigation of the proportion of difficult choices selected by participants in relation to the amount of reward, with the factors Exercise (cycling, no-cycling) and amount (1.5, 3, 4.5, 6). Furthermore, the proportion of "hard" choices selected by the participants was examined in relation to the probability of reward, with the factors Exercise (cycling, no-cycling) and probability (10%, 50%, 90%). Pearson correlation was employed to assess the relationship between the percentage of difficult choices at each probability and the total amount of money won by participants.

In order to ascertain whether the P300 could serve as a neural marker of motivation, Pearson correlation was employed to assess the relationship between the number of difficult choices at 10% and the amplitude of the P300, in accordance with the findings of [21].

**Scales and questionnaires.** Scales and questionnaires were employed to gauge the perceived exertion of the subjects during the EEfRT. A one-way repeated measures ANOVA was conducted to compare the perceived exertion reported on the BORG scale after the training period and at the conclusion of each training block.

To ascertain whether there were any differences in mood, a general linear model was employed, with BMIS scores treated as repeated measures and the factors Exercise (cycling, no-cycling) and Time of Realisation (before, after) included as variables. BMIS scores comprised the Pleasant-Unpleasant and Aroused-Calm scales. Pearson correlations were used to assess the relationship between BMIS score and P300 or FRN amplitudes. Spearman correlations were used to assess the relationship between the hour of the day the subjects participated in the experiment, BMIS scores, and the number of difficult choices made in the EEfRT.

## Results

Table 1 provides a summary of the characteristics of the participants in this study. The median HR at rest was 70 beats per minute (bpm), with a mean HR target of 133 bpm. The median power was set at 80 W. Six out of the 20 participants were categorised in the HEPA (health-enhancing physical activity) active group, for individuals who exceed the minimum public health physical activity recommendations. Thirteen participants were in the minimally active group, who are sufficiently active. One participant was in the inactive group, representing the lowest level of physical activity and individuals considered as insufficiently active.

**Table 1. Participants' characteristics. IPAQ total score is in MET-min per week.**

| Subject | Age | Sex | HR rest (bpm) | HR Target (bpm) | Power (W) | IPAQ | | Mean Borg rating |
| --- | --- | --- | --- | --- | --- | --- | --- | --- |
| | | | | | | Total Score (MET-min/week) | Ranking | |
| 1 | 24 | F | 67 | 132 | 80 | 2118 | Minimally active | 11,6 |
| 2 | 26 | M | 65 | 130 | 90 | 2481 | HEPA active | 13,4 |
| 3 | 24 | F | 91 | 144 | 70 | 1880 | HEPA active | 11,6 |
| 4 | 23 | M | 81 | 139 | 90 | 2739 | Minimally active | 13,6 |
| 5 | 24 | F | 80 | 139 | 40 | 745 | Minimally active | 14,4 |
| 6 | 24 | F | 72 | 134 | 60 | 1746 | Minimally active | 11,2 |
| 7 | 22 | F | 80 | 139 | 60 | 576 | inactive | 13,4 |
| 8 | 24 | M | 56 | 126 | 90 | 1609 | Minimally active | 13 |
| 9 | 21 | M | 70 | 135 | 100 | 4998 | HEPA active | 12,8 |
| 10 | 23 | F | 62 | 130 | 80 | 2982 | Minimally active | 13,6 |
| 11 | 30 | M | 46 | 118 | 100 | 1493 | Minimally active | 14 |
| 12 | 34 | F | 65 | 126 | 80 | 6079 | HEPA active | 15,2 |
| 13 | 40 | F | 65 | 123 | 50 | 2490 | Minimally active | 11,4 |
| 14 | 27 | F | 72 | 132 | 60 | 2910 | Minimally active | 9,8 |
| 15 | 24 | F | 73 | 135 | 60 | 1260 | Minimally active | 13,2 |
| 16 | 20 | F | 80 | 140 | 60 | 872 | Minimally active | 13,4 |
| 17 | 23 | M | 70 | 134 | 90 | 2250 | Minimally active | 15,4 |
| 18 | 28 | F | 64 | 128 | 100 | 5253 | HEPA active | 11,2 |
| 19 | 30 | F | 52 | 121 | 80 | 1878 | HEPA active | 9,8 |
| 20 | 26 | F | 77 | 139 | 50 | 835 | Minimally active | 14,6 |

## EEG results at the EEfRT

The results demonstrated the presence of clear ERPs in both the cycling and no-cycling sessions (Fig 4). The results demonstrated that the amplitude of the P300 was influenced by the cycling condition, irrespective of feedback or electrodes (F(1,18) = 4.63; p = 0.045; partial eta-squared = 0.20). The amplitude was observed to be higher in the no-cycling session. However, no significant effect of the cycling condition was found on the amplitude of the FRN.

A significant difference in P300 amplitude was observed between reward and no-reward feedback conditions, with a laterality effect (F(2,36) = 3.70; p = 0.035; partial eta-squared = 0.17). The amplitude was significantly higher for the left (p = 0.016) and right (p = 0.036) sides. In these regions, the amplitude was observed to be higher following the presentation of a reward. Furthermore, the feedback had an impact on the amplitude of the FRN, according to anteriority (F(2,36) = 5.74; p = 0.007; partial eta-squared = 0.24), with significant differences observed on the posterior line (p = 0.005). In this region, a smaller response was observed following the presentation of a reward.

No significant effect of session order on the amplitude of the P300 or the FRN was identified.

## Behavioral results at the EEfRT

The proportion of hard choices was significantly higher for high probabilities than for low probabilities of gain (Fig 5a) (F(2,38)=73.194; p<0.0001), with significant differences between all probability levels (p<0.0001 for all). The proportions of hard choices were also higher when the amount of points at stake were higher (Fig 5b) (F(3,57)=66.59; p<0.0001). A

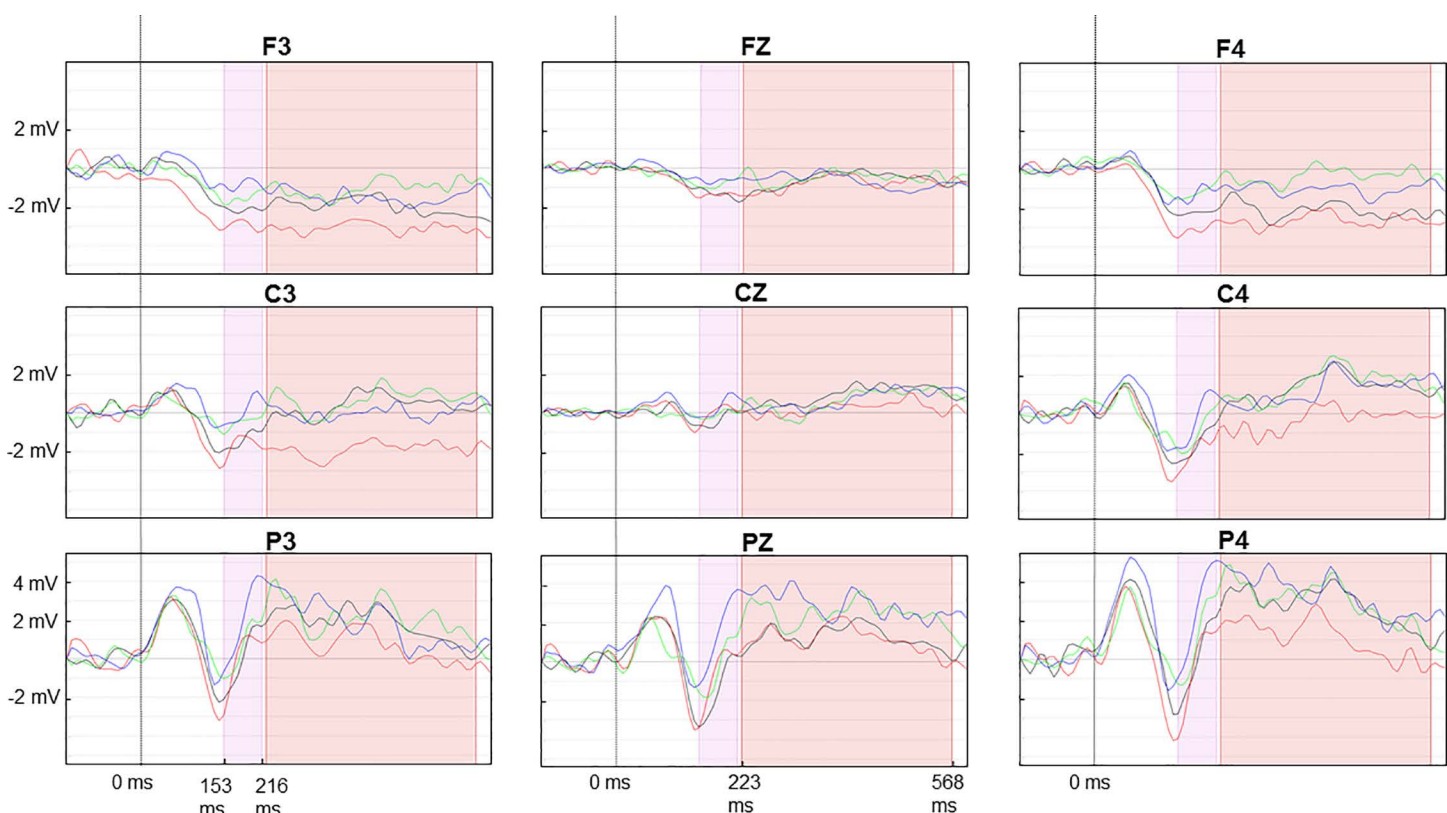

**Fig 4. Visualization of the mean ERPs for the 9 electrodes included in the analysis.** Pink time-window: 153 to 216 ms (early reward processing), Red time-window: 223 to 568 ms (reward processing); Blue curve condition: no cycling, no reward; Green curve: no cycling, rewards; Red curve: cycling, no reward; Black curve: cycling, rewards.

significantly lower proportion of difficult choices was observed at 1.5 points in comparison to all other amounts (p<0.0001 for all), as well as between 3 and 6 points (p<0.0001).

The data revealed that the proportion of hard choices was similar in both the cycling and no-cycling conditions, with no significant differences in the proportion of difficult choices among probabilities and among the amount of points at stake. Furthermore, there were no significant differences in the final score between the cycling and no-cycling conditions.

The total number of points accumulated by the participants was found to be significantly correlated with the percentage of difficult choices at 10% (r=0.62; p=0.003), 50% (r=0.94; p<0.00001) and 90% (r=0.50; p=0.03) in the no-cycling condition. Additionally, a correlation was observed between the total points earned by the participants while cycling and the percentage of difficult choices at 10% (r=0.72; p=0.0003), 50% (r=0.85; p<0.00001) and 90% (r=0.54; p=0.01).

A comparison of the relationship between behavioural and neural responses at the EEfRT revealed no significant correlation between the amplitude of P300 at Cz and the percentage of difficult choices at 10% in both the cycling and no-cycling sessions. Nevertheless, a notable correlation was identified at the Pz electrode between the number of challenging selections at 10% and the amplitude of the P300 following a favourable outcome (r=0.47; p=0.04) during the cycling phase.

## Scale and questionnaire

The Borg rating of perceived exertion was employed to quantify the level of physical exertion experienced by the participants. The Borg scale was used to assess perceived exertion, which was found to increase significantly during the task (F(4,76) = 19.65, p<0.00001). The perceived effort prior to the initial block was significantly less than that observed during the performance of the task (p<0.0001 for all). During the task, no difference was observed between the different blocks (Fig 6). No relationship was identified between the mean Borg rating and the IPAQ score.

BMIS scores were similar at the beginning of both the cycling and no-cycling sessions. There were no significant differences in BMIS scores after the cycling and no-cycling sessions. Furthermore, the EEfRT did not influence BMIS scores during the task.

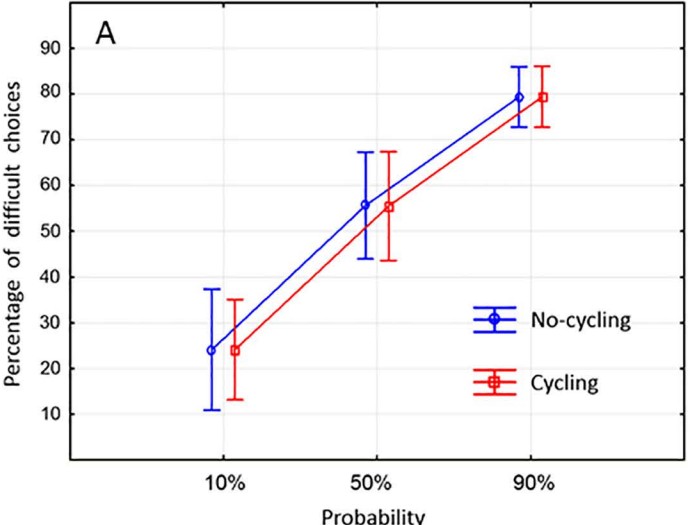 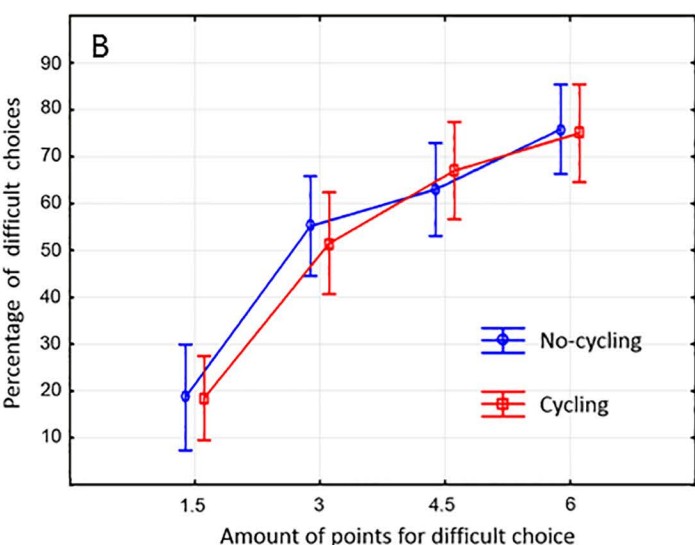

**Fig 5. A. Percentages of difficult choices according to probability of reward when the trial was completed. B. Percentages of difficult choices according to possible rewards if the trial is completed.**

## Correlations

No correlation was identified between P300 or FRN amplitude and either of the BMIS scores. We also examined whether the time of the experiment could influence emotional state and cognitive performance. A relationship was found between the hour of the day at which participants participated in the experiment and BMIS scores, but only in the no-cycling condition ($rs = 0.51$, $p = 0.02$ for the pleasant-unpleasant scale; $rs = 0.50$, $p = 0.02$ for the aroused-calm scale). No correlation was found between the hour of the day at which participants participated in the experiment and the number of difficult choices made in the EEfRT.

## Discussion

The objective of the present study was to assess the feasibility of measuring objective markers of motivation during cycling. Subjects participated in an effort-based decision-making task, the Effort Expenditure Ratio Task (EEfRT), while either cycling or at rest. EEG was used to record their cerebral activity. The EEfRT is a relevant paradigm that has been demonstrated to elicit a range of ERPs associated with motivation and decision-making processes [21]. In this study, we focused on examining the brain responses associated with the reward prediction error, which occurs when the feedback or result differs from the expected outcome. Our findings revealed a correlation between the P300 and the proportion of challenging choices made by participants during cycling, suggesting that this ERP may serve as a neural marker of motivation.

The ERP wave was observed in both the cycling and non-cycling conditions, thereby confirming the feasibility of recording cerebral activity during cycling [26]. The amplitude of the P300 wave was nevertheless observed to be reduced during the cycling session. The impact of cycling on ERPs, in comparison to a scenario in which ERPs are recorded without cycling, is a topic of considerable debate. Some studies have indicated that the P300 wave amplitude is greater in the cycling condition than in the resting condition, while other studies have reported the opposite or no differences (for a review, see [26]). The discrepancies across studies can be attributed to a number of factors, notably the considerable variability observed in terms of intensity and duration. The paradigm employed to elicit the P300 is also a significant factor, as the majority of previous studies measuring ERPs in response to cycling have utilized attentional tasks [34]. In this study, the task was a more complex decision-making task that required not only attentional resources but also cognitive resources to evaluate the motivation to exert effort [21]. The reduction in P300 amplitude indicates a shift in cognitive activity in response to exercise, in line with the transient hypofrontality theory [35] and the reticular-activating hypofrontality model [36]. These theories posit that cognitive resources are transferred from brain structures such as the prefrontal cortex to regions that require them for physical effort due to resource limitations in the body. The discrepancy in findings

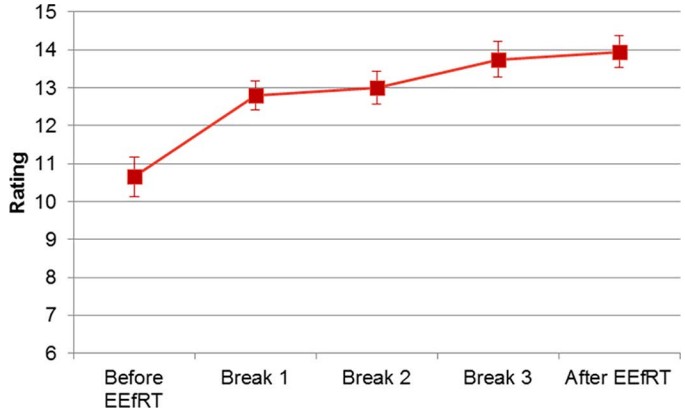

**Fig 6. Evolution of Borg rating during the EEfRT.**

between the P300 and FRN may be attributed to the low level of cognitive demand in early reward processes, whereby transient cognitive resource limitations have no effect on early reward processing. While it cannot be ruled out that the observed amplitude differences are residual pedalling-related artefacts, one study found that there was no correlation between muscle artefacts and P300 amplitude [37].

In addition to the impact of cycling or non-cycling conditions, the amplitude of the FRN and the P300 components were influenced by the feedback outcome (rewards versus no-rewards), thereby confirming their role in reward prediction [38]. The amplitude of the P300 wave was observed to be greater for rewards than for no-rewards in the central line (C3, Cz, C4). This finding is consistent with previous research that has demonstrated an altered amplitude on centroparietal electrodes [18,21,39]. The FRN exhibited a higher amplitude in the absence of rewards compared to rewards for the parietal line (P3, Pz, P4) irrespective of the cycling condition. Notably, this was particularly evident at the Pz electrode, which aligns with the observations made by [21] on the same paradigm but at more central electrodes. The reward prediction error signal is a marker of expectancy violation, generated when an outcome is worse than expected. As this signal of discrepancy is hypothesised to be used to modify behaviour functionally [40], the neural differences observed between rewards and no-rewards indicate that the participants were aware of the task, regardless of whether they were cycling.

The discrepancy in P300 amplitude between the cycling and no-cycling conditions was not reflected in the participants' behaviour, which remained largely consistent. The behavioural results pertaining to the choices made are of interest, as the proportion of difficult choices made in specific gain and probability conditions may serve as an indicator of motivation related to the anticipation of reward [22]. In the present study, participants elected to undertake the challenging task with greater frequency when the potential reward or probability of reward associated with the challenging task was higher. The strategy remained consistent across both conditions, indicating that the decision-making process was not influenced by the cycling. In a separate study, Wang and colleagues employed the Wisconsin Card Sorting Task (WCST) as a decision-making task, without cycling (control) and during cycling at 30% (light intensity), 50% (moderate intensity) and 80% (high intensity) of heart rate reserve for approximately 20 minutes [41]. The WCST is employed for the assessment of mental flexibility, which is a component of executive function. Performance was found to be reduced solely in the case of the high-intensity condition (80% of HR reserve), with no impact observed in the other conditions. In the present study, the exercise was of moderate intensity (50% of HR reserve) and lasted for approximately 20 minutes (without including pause times), which is comparable to the moderate condition described by Wang and colleagues. Despite an increase in reported scores of perceived effort between the training session and the EEfRT, these scores remained constant throughout the task, indicating that the effort was not a significant factor. Previous research has demonstrated that cognitive performance may not be significantly affected by short durations and light to moderate intensities of exercise [42]. It would be interesting to evaluate whether the participants' behaviour in effort-based choices would be impacted by an exercise of higher intensity.

In relation to these behavioural results, previous studies have demonstrated that the P300 can be used as a measure of participant engagement in the task. Furthermore, this ERP has been shown to be correlated with the proportion of difficult choices at the low probability [21]. This condition is deemed to be the most appropriate for reflecting the strength of motivation. Furthermore, a correlation was identified in the cycling condition, indicating that the P300 may serve as a neural marker of motivation, in addition to reflecting outcome processing.

Furthermore, given the established influence of mood on decision-making [30], we also evaluated the impact of cycling on reported mood states. The aim was to assess whether mood could be related to effort-based decision-making, and consequently on the strength of motivation. The results indicated no significant effect of exercise on BMIS scores. This finding aligns with the results of a study that compared BMIS scoring in participants engaged in sitting sessions and those engaged in walking sessions [43]. It is plausible that the intensity of exercise plays a pivotal role in influencing both mood and decision-making. Consequently, further research is necessary to investigate this hypothesis.

It should be noted that this study is subject to a number of limitations. The inclusion criteria were not as strictly defined as they might otherwise have been, particularly in relation to gender and/or sporting practice. The level of sporting activity may be a factor that could influence behaviour. In the study conducted by Wang and colleagues [41], all participants were students from a university with a strong emphasis on sports, resulting in a cohort that was predominantly young and engaged in regular physical activity. In our study, the participants were approximately the same age but exhibited a greater range of physical conditions. According to the IPAQ, only six out of 20 participants were classified as being in the high level of activity category. It has been demonstrated that an individual's fitness level may serve to mitigate the adverse effects of exercise on executive neural functions [44]. The impact of acute physical exercise on executive functions may vary according to age [45]. The participants in this study were young adults with a range of physical exercise habits. It would be beneficial to replicate this study in an older sample in order to gain a deeper understanding of motivation during exercise, particularly given the intention to evaluate motivational deficits during physical rehabilitation, which predominantly involves patients over the age of 50. Furthermore, the time of the day is known to impact Heart Rate Variability [46]. Here, the experiment was conducted at different times in the afternoon, which may have resulted in varying resting heart rates among participants [46]. Moreover, a relationship was observed between the time of the afternoon at which subjects participated in the experiment and mood scores, but only in the no-cycling condition, making the results difficult to interpret. This variability is somewhat mitigated by the crossover design, in which each individual serves as their own control, eliminating inter-subject variability in group comparisons and reducing the influence of covariates [47]. This design entailed participants completing the EEfRT twice, once while cycling and once without. Consequently, in the second session, the participants were already acquainted with the task and had already formulated a strategy. Nevertheless, the behavioural outcomes remained consistent, and the order of the sessions had no impact on the ERPs. A number of participants proactively disclosed their unease with the cycloergometer, particularly in the absence of cycling. It is possible that the condition of being in the no-cycling state may have induced head agitation, which in turn may have caused artefacts to be produced. It should be noted that no remuneration was offered to the participants. It is established that the presence of a monetary incentive has an impact on motivation [48]. Nevertheless, a study was able to successfully measure a reward-positive ERP (occurring approximately 250–350 ms post-feedback) in a non-monetary gambling task, with higher amplitudes for positive feedback relative to negative feedback [49]. The proportion of difficult choices in this study was found to be no different from that observed in the study by [21], in which monetary incentive was present.

## Conclusion

The results of this feasibility study demonstrated the viability of measuring motivation-related event-related potentials (ERPs) from an effort-based decision-making task while engaging in physical exercise. Additionally, the variability in EEG measures helped refine the sample size estimate for the full-scale study, ensuring it is adequately powered to detect meaningful effects. To ensure adequate statistical power for detecting differences in event-related potentials (ERPs) between cycling and no-cycling conditions, with an effect size of $d = 0.2$, a significance level of $\alpha = 0.05$, and a power of 0.80, a required sample size of 50 participants will be necessary. Moreover, further research is required to adapt the design to enable the measurement of neural markers of motivation related to the physical exercise of cycling itself. This will entail an adaptation of the type of exercise used to measure motivation, whether aerobic or not, with the aim of increasing the pleasure derived from performing the task and also to enable exercise to be undertaken autonomously at home. Once this adaptation has been made, populations in rehabilitation will be evaluated in order to gain a better understanding of their willingness to engage in physical activity.

## Author contributions

**Conceptualization:** Damien Gabriel, Laurent Mourot, Julie Giustiniani, Lionel Pazart.

**Data curation:** Rémi Renoud-Grappin, Emmanuelle Broussard.

**Formal analysis:** Rémi Renoud-Grappin, Emmanuelle Broussard, Lionel Pazart.

**Funding acquisition:** Lionel Pazart.

**Investigation:** Rémi Renoud-Grappin, Emmanuelle Broussard.

**Methodology:** Rémi Renoud-Grappin, Damien Gabriel, Laurent Mourot, Julie Giustiniani, Lionel Pazart.

**Project administration:** Rémi Renoud-Grappin, Lionel Pazart.

**Resources:** Lionel Pazart.

**Supervision:** Damien Gabriel, Julie Giustiniani, Lionel Pazart.

**Validation:** Damien Gabriel, Lionel Pazart.

**Visualization:** Rémi Renoud-Grappin.

**Writing – original draft:** Rémi Renoud-Grappin, Damien Gabriel, Lionel Pazart.

**Writing – review & editing:** Rémi Renoud-Grappin, Damien Gabriel, Laurent Mourot, Julie Giustiniani, Lionel Pazart.

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
