## [Decision Letter · Decision Letter 0]

18 Mar 2025

PONE-D-24-46303A Measure of Event-Related Potentials (ERP) Indices of Motivation During CyclingPLOS ONE

Dear Dr. Gabriel,

Thank you for submitting your manuscript to PLOS ONE. After careful consideration, we feel that it has merit but does not fully meet PLOS ONE’s publication criteria as it currently stands. Therefore, we invite you to submit a revised version of the manuscript that addresses the points raised during the review process. Both reviewers are positive with the quality of the submitted study. However, one reviewer raised minor comments that should be addressed. I invite the authors to address the suggested minor changes as soon as possible and resubmit for acceptance. Please submit your revised manuscript by May 02 2025 11:59PM. If you will need more time than this to complete your revisions, please reply to this message or contact the journal office at plosone@plos.org . Please include the following items when submitting your revised manuscript:

We look forward to receiving your revised manuscript.

Kind regards,

Vilfredo De Pascalis

Academic Editor

PLOS ONE

Journal Requirements:

Additional Editor Comments :

Both reviewers are positive with the quality of the submitted study. However, one reviewer raised minor comments that should be addressed. I invite the authors to address the suggested minor changes as soon as possible and resubmit for acceptance.

Reviewers' comments:

Reviewer's Responses to Questions

**Comments to the Author**

1. Is the manuscript technically sound, and do the data support the conclusions?

Reviewer #1: Yes

Reviewer #2: Yes

2. Has the statistical analysis been performed appropriately and rigorously? 

Reviewer #1: Yes

Reviewer #2: Yes

3. Have the authors made all data underlying the findings in their manuscript fully available?

Reviewer #1: Yes

Reviewer #2: Yes

4. Is the manuscript presented in an intelligible fashion and written in standard English?

Reviewer #1: Yes

Reviewer #2: Yes

5. Review Comments to the Author

Reviewer #1: This study is very interesting. The participants were required to complete two distinct sessions: one in which they did not engage in cycling, and the latter session was conducted on a cycloergometer. The authors found that the P300 amplitude was influenced by the act of cycling, exhibiting a reduction during the cycling session. A positive correlation was identified between P300 amplitude and the proportion of difficult choices, particularly under conditions of low reward probability. The热for, this artical suggests that P300 may serve as a neural marker of motivation.

Reviewer #2: Dear Editor,

Thank you very much for inviting me to review this manuscript. I appreciate the opportunity to contribute to the evaluation of this study. This study has significant value in terms of methodological innovation and practical application potential, but further improvement is needed in terms of experimental design rigor, depth of result interpretation, and literature support. Below are my detailed comments and suggestions for the authors' consideration:

Q1: Study Design Limitations

The study design has some limitations that should be addressed. The sample size is relatively small, with only 20 participants. This may restrict the generalizability and external validity of the findings.

Suggestion: It would be beneficial to include a power analysis to determine whether the number of studies included is adequate for drawing strong conclusions.

Q2: The potential impact of variables on the results

In the discussion, the authors mention that "some studies have shown that the amplitude of the P300 wave is greater in the cycling condition than in the resting condition, while other studies have reported the opposite or no difference." It is unclear whether the resting conditions in these other studies are comparable to the one used in this study. The authors have noted that sitting on a bicycle without cycling may induce a unique psychological effect. Despite the use of a crossover design, potential confounding variables at different time points (e.g., environmental noise, differences in participants' psychological states) and individual states (e.g., fatigue levels, circadian rhythms) may vary. These factors could influence participants' emotions, cognition, and physiological responses, thereby affecting the study results.

Suggestion: The authors should provide additional information on the control methods for environmental variables and psychological states in the experimental design or discuss the potential impact of these factors on the results.

Q3: High-Resolution Presentation of the Experimental Procedure

If possible, please provide a high-resolution version of the experimental procedure (i.e., Figure 3). This would facilitate better understanding of the study for other readers.

Q4: Insufficient Details in EEG Data Processing

The manuscript mentions the use of ICA to remove eyeblink artifacts but does not specify how movement artifacts (e.g., electromyographic interference) caused by cycling were handled.

Suggestion: Please provide more detailed information on the handling of movement artifacts in the EEG data processing section.

Q5: Clarity and Conciseness of Language and Expression

Some paragraphs are rather lengthy and lack clarity in logic, which may hinder readers' comprehension. For example, the description of the EEfRT task could be more concise, avoiding redundancy and repetition.

Suggestion: Further language refinement is recommended to simplify long sentences and enhance logical coherence.

Q6: Insufficient Depth in Analysis of Limitations

While the study has identified some limitations, the analysis of certain limitations is not sufficiently in-depth. For example, it mentions that conducting the experiment at different times may affect the results but does not further explore the specific mechanisms by which time factors influence resting heart rate, emotional state, and cognitive performance.

Suggestion: It is suggested that each limitation be analyzed more thoroughly to explore its potential multifaceted impacts on the study results.

Q7: Insufficient Update of References

Some of the references are outdated and may not reflect the latest research developments in the field. For example, "Mayer,John D.,et Yvonne N.Gaschke. 1988", "Oldfield, R.C. 1971”, “Quattrone, George A. 1985”, and “Sutton, S.,M.Braren, J.Zubin, et E.R.John. 1965”, etc.

Suggestion: It is recommended to update the references to include the latest (within the past 5 years) literature in order to enhance the timeliness and scientific validity of the research.

Overall, the study provides valuable insights into the use of EEG to measure motivation during cycling. However, addressing the above points would significantly enhance the robustness and clarity of the research. I look forward to seeing the revised manuscript.

6. PLOS authors have the option to publish the peer review history of their article (what does this mean? ). If published, this will include your full peer review and any attached files.

**Do you want your identity to be public for this peer review?** For information about this choice, including consent withdrawal, please see our Privacy Policy .

Reviewer #1: **Yes: ** shidong Yang

Reviewer #2: No

---

## [Author Response · Author response to Decision Letter 1]

28 Mar 2025

First, we would like to thank the reviewers for their precious comments.

Q1: Study Design Limitations

The study design has some limitations that should be addressed. The sample size is relatively small, with only 20 participants. This may restrict the generalizability and external validity of the findings.

Suggestion: It would be beneficial to include a power analysis to determine whether the number of studies included is adequate for drawing strong conclusions.

Response: We agree with the reviewer's comment. Regarding the generalizability of the findings, partial eta squared values are presented along with p values relative to FRN and P300. Moreover, one of the goals of this feasibility study was to help refining the sample size estimate for a full-scale study. We added the following text in the conclusion:

"The results of this feasibility study demonstrated the viability of measuring motivation-related event-related potentials (ERPs) from an effort-based decision-making task while engaging in physical exercise. Additionally, the variability in EEG measures helped refine the sample size estimate for the full-scale study, ensuring it is adequately powered to detect meaningful effects. To ensure adequate statistical power for detecting differences in event-related potentials (ERPs) between cycling and no-cycling conditions, with an effect size of d = 0.2, a significance level of α = 0.05, and a power of 0.80, a required sample size of 50 participants will be necessary."

Q2: The potential impact of variables on the results

In the discussion, the authors mention that "some studies have shown that the amplitude of the P300 wave is greater in the cycling condition than in the resting condition, while other studies have reported the opposite or no difference." It is unclear whether the resting conditions in these other studies are comparable to the one used in this study. The authors have noted that sitting on a bicycle without cycling may induce a unique psychological effect. Despite the use of a crossover design, potential confounding variables at different time points (e.g., environmental noise, differences in participants' psychological states) and individual states (e.g., fatigue levels, circadian rhythms) may vary. These factors could influence participants' emotions, cognition, and physiological responses, thereby affecting the study results.

Suggestion: The authors should provide additional information on the control methods for environmental variables and psychological states in the experimental design or discuss the potential impact of these factors on the results.

Response: We added additional information on how we controlled some environmental variables, especially the time of the day, and the interval between sessions:

"The order of the sessions was randomly assigned to each participant and both sessions took place in the afternoon. The interval between two sessions was kept as short as possible, depending on the participant's availability, in order to avoid variations in their psychological state as well as variations in the external environment (e.g. temperature)."

Moreover, the inclusion criteria were also designed to reduce variations in emotional state that could affect the EEfRT:

"Since anhedonia, depression or other psychiatric conditions have been shown to impact EEfRT performance (Treadway et al. 2009), individuals with a psychiatric history were excluded from this study."

We also modified a sentence about the time of day in the discussion, which was initially confusing:

" Furthermore, the time of the day is known to impact Heart Rate Variability (Choi et al., 2008). Here, the experiment was conducted at different times in the afternoon, which may have resulted in varying resting heart rates among participants (Choi et al., 2008)"

Q3: High-Resolution Presentation of the Experimental Procedure

If possible, please provide a high-resolution version of the experimental procedure (i.e., Figure 3). This would facilitate better understanding of the study for other readers.

Response: We have recreated Figure 3 so that it has better resolution

Q4: Insufficient Details in EEG Data Processing

The manuscript mentions the use of ICA to remove eyeblink artifacts but does not specify how movement artifacts (e.g., electromyographic interference) caused by cycling were handled.

Suggestion: Please provide more detailed information on the handling of movement artifacts in the EEG data processing section.

Response: Some additional information has been added in the methods section: "To remove artifacts caused by movement, and any other non-physiological factors, EEG epochs were cleaned with a visual trial-by-trial inspection."

We also discuss the possible impact of muscular artifacts in the discussion:

"While it cannot be ruled out that the observed amplitude differences are residual pedalling-related artefacts, one study found that there was no correlation between muscle artefacts and P300 amplitude (Zink et al. 2016)."

Q5: Clarity and Conciseness of Language and Expression

Some paragraphs are rather lengthy and lack clarity in logic, which may hinder readers' comprehension. For example, the description of the EEfRT task could be more concise, avoiding redundancy and repetition.

Suggestion: Further language refinement is recommended to simplify long sentences and enhance logical coherence.

Response:

We tried as much as possible to simplify the description of the EEfRT:

"The EEfRT, adapted from Treadway et al. (2009) for ERP analysis (Giustiniani et al., 2020; Figure 3), aims to maximize points by completing easy or hard tasks. Task selection depends on the potential reward and the probability of receiving it.

In this study, the task included 120 trials. The easy task required 70% of the participant’s calibrated maximum button presses with the right index finger within 7 seconds, awarding 1 point upon success. The hard task required 90% of the maximum presses using the left-hand auricular finger within 14 seconds, earning 1.5, 3, 4.5, or 6 points. The hard task’s time limit was reduced from 21 to 14 seconds to accommodate the increased trial count. Reward probabilities (10%, 50%, or 90%) applied equally to both tasks.

Figure 3 outlines the trial sequence:

-A 1-second screen displayed the reward probability and point values for easy (1 point) and hard (variable points) tasks.

-A circle appeared, prompting task selection.

-Participants pressed buttons to fill a bar before time expired.

-Success was indicated by a cross (1 s) followed by feedback (2 s): a green square for a win, a red square for a loss. Failure resulted in a red square for 2 seconds."

Q6: Insufficient Depth in Analysis of Limitations

While the study has identified some limitations, the analysis of certain limitations is not sufficiently in-depth. For example, it mentions that conducting the experiment at different times may affect the results but does not further explore the specific mechanisms by which time factors influence resting heart rate, emotional state, and cognitive performance.

Suggestion: It is suggested that each limitation be analyzed more thoroughly to explore its potential multifaceted impacts on the study results.

Response:

Following the reviewer's suggestion, we compared the emotional state of all participants at the beginning of the cycling and no-cycling sessions. No difference was found:

"BMIS scores were similar at the beginning of both the cycling and no-cycling sessions."

Regarding the experiment that took place at different times of the day, a reanalysis of our data showed that no recording took place in the morning. 4 of the 40 recordings took place early in the afternoon and 36 in the middle/late afternoon. Finally, there is not much difference and the time of the recordings are too similar to compare the data in groups.

We nevertheless used correlations measures to explore the mechanisms by which time factors influence mood and scores at the EEfRT. We added the following text in the methods section:

“Spearman correlations were used to assess the relationship between the hour of the day the subjects participated in the experiment, BMIS scores, and the number of difficult choices made in the EEfRT.”

In the results section:

“We also examined whether the time of the experiment could influence emotional state and cognitive performance. A relationship was found between the hour of the day at which participants participated in the experiment and BMIS scores, but only in the no-cycling condition (rs = 0.51, p = 0.02 for the pleasant-unpleasant scale; rs = 0.50, p = 0.02 for the aroused-calm scale). No correlation was found between the hour of the day at which participants participated in the experiment and the number of difficult choices made in the EEfRT.”

And in the discussion:

" Furthermore, the time of the day is known to impact Heart Rate Variability (Choi et al., 2008). Here, the experiment was conducted at different times in the afternoon, which may have resulted in varying resting heart rates among participants (Choi et al., 2008). Moreover, a relationship was observed between the time of the afternoon at which subjects participated in the experiment and mood scores, but only in the no-cycling condition, making the results difficult to interpret."

Q7: Insufficient Update of References

Some of the references are outdated and may not reflect the latest research developments in the field. For example, "Mayer,John D.,et Yvonne N.Gaschke. 1988", "Oldfield, R.C. 1971”, “Quattrone, George A. 1985”, and “Sutton, S.,M.Braren, J.Zubin, et E.R.John. 1965”, etc.Suggestion: It is recommended to update the references to include the latest (within the past 5 years) literature in order to enhance the timeliness and scientific validity of the research.

Response: We understand the importance of incorporating up-to-date literature.

However, some of the references in our manuscript, such as Oldfield (1971), are foundational and remain the standard citations for specific concepts, tools, or methodologies (e.g., the Edinburgh Handedness Questionnaire or the BMIS). As these works are still widely referenced and have not been replaced by more recent alternatives, they are necessary to provide historical and methodological context. It is the same with Mayer and Gaschke 1988, which is the reference article for the BMIS.

That said, we recognize the value of including recent studies. We changed the reference of Quattrone 1985 to Dang et al., 2020. Regarding the P300, Sutton is the pioneer study, but we also added a more recent reference: Grasso-Cladera et al., 2024.

---

## [Decision Letter · Decision Letter 1]

22 Apr 2025

A Measure of Event-Related Potentials (ERP) Indices of Motivation During Cycling

PONE-D-24-46303R1

Dear Dr. Gabriel,

We’re pleased to inform you that your manuscript has been judged scientifically suitable for publication and will be formally accepted for publication once it meets all outstanding technical requirements.

Kind regards,

Vilfredo De Pascalis

Academic Editor

PLOS ONE

Additional Editor Comments (optional):

Considering that all comments have been properly addressed, we are glad to accept the manuscript for publication.

Reviewers' comments:

Reviewer's Responses to Questions

**Comments to the Author**

1. If the authors have adequately addressed your comments raised in a previous round of review and you feel that this manuscript is now acceptable for publication, you may indicate that here to bypass the “Comments to the Author” section, enter your conflict of interest statement in the “Confidential to Editor” section, and submit your "Accept" recommendation.

Reviewer #2: All comments have been addressed

2. Is the manuscript technically sound, and do the data support the conclusions?

Reviewer #2: Yes

3. Has the statistical analysis been performed appropriately and rigorously? 

Reviewer #2: Yes

4. Have the authors made all data underlying the findings in their manuscript fully available?

Reviewer #2: Yes

5. Is the manuscript presented in an intelligible fashion and written in standard English?

Reviewer #2: Yes

6. Review Comments to the Author

Reviewer #2: First and foremost, I would like to extend my sincere appreciation to the editorial team for entrusting me with the responsibility of reviewing this manuscript. It has been an honor to contribute to the peer-review process of such a respected journal.

Having carefully reviewed the authors' responses to my previous comments, I am pleased to report that the revisions submitted are of exceptional quality. The authors have demonstrated remarkable diligence in addressing each concern with thoroughness and scientific rigor.

Given the thoroughness of the revisions and the authors' responsiveness to feedback, I am confident that this manuscript now meets the rigorous standards required for publication in your esteemed journal.

Once again, I express my deepest gratitude to the editorial team for facilitating this collaborative process, and to the authors for their dedication to scientific excellence. I look forward to the opportunity to contribute further to the journal's peer-review process in the future.

7. PLOS authors have the option to publish the peer review history of their article (what does this mean? ). If published, this will include your full peer review and any attached files.

**Do you want your identity to be public for this peer review?** For information about this choice, including consent withdrawal, please see our Privacy Policy .

Reviewer #2: **Yes: ** Jiayue Cui

---

## [Editor Report · Acceptance letter]

PONE-D-24-46303R1

PLOS ONE

Dear Dr. Gabriel,

I'm pleased to inform you that your manuscript has been deemed suitable for publication in PLOS ONE. Congratulations! Your manuscript is now being handed over to our production team.

Kind regards,

on behalf of

Prof. Vilfredo De Pascalis

Academic Editor

PLOS ONE